# Identification and Localisation Algorithm for Sugarcane Stem Nodes by Combining YOLOv3 and Traditional Methods of Computer Vision

**DOI:** 10.3390/s22218266

**Published:** 2022-10-28

**Authors:** Deqiang Zhou, Wenbo Zhao, Yanxiang Chen, Qiuju Zhang, Ganran Deng, Fengguang He

**Affiliations:** 1School of Mechanical Engineering, Jiangnan University, Wuxi 214000, China; 2Agro-Machinery Research Institute, Chinese Academy of Tropical Agricultural Sciences, Zhanjiang 524000, China

**Keywords:** automatic cutting, sugarcane stem nodes, YOLOv3, preprocess, improved edge extraction algorithm

## Abstract

Sugarcane stem node identification is the core technology required for the intelligence and mechanization of the sugarcane industry. However, detecting stem nodes quickly and accurately is still a significant challenge. In this paper, in order to solve this problem, a new algorithm combining YOLOv3 and traditional methods of computer vision is proposed, which can improve the identification rate during automated cutting. First, the input image is preprocessed, during which affine transformation is used to correct the posture of the sugarcane and a rotation matrix is established to obtain the region of interest of the sugarcane. Then, a dataset is built to train the YOLOv3 network model and the position of the stem nodes is initially determined using the YOLOv3 model. Finally, the position of the stem nodes is further located accurately. In this step, a new gradient operator is proposed to extract the edge of the image after YOLOv3 recognition. Then, a local threshold determination method is proposed, which is used to binarize the image after edge extraction. Finally, a localization algorithm for stem nodes is designed to accurately determine the number and location of the stem nodes. The experimental results show that the precision rate, recall rate, and harmonic mean of the stem node recognition algorithm in this paper are 99.68%, 100%, and 99.84%, respectively. Compared to the YOLOv3 network, the precision rate and the harmonic mean are improved by 2.28% and 1.13%, respectively. Compared to other methods introduced in this paper, this algorithm has the highest recognition rate.

## 1. Introduction

Sugarcane is the main sugar crop, and sugar is an important agricultural product in China. The sugarcane industry provides support for economic development and the increment of farmers’ income. Mechanization and refinement of the whole sugarcane planting process is a trend of industrial development, but most sugarcane seed cutting machines in China do not have an anti-injury function to prevent damage to buds during the automatic sugarcane seed cutting process, which restricts the development of the sugarcane industry [1,2].

At present, computer vision detection systems developed for agriculture are widely used in fruit classification, grain classification, food identification, weed detection, medicinal plants, and other fields [3,4,5,6]. The development of machine vision has provided novel technical means for automation of agricultural production. Research and development of a sugarcane bud anti-injury system based on machine vision is still in its infancy worldwide. Nare developed an automated production system for sugarcane seeds [7]. The system integrated the processes of mechanical feeding, cutting, and separating mechanisms with an electronic control system. In a previous study of ours, a sugarcane seed cutting system based on machine vision was designed. A gradient feature vector was designed to calculate the column gradient value and a location search algorithm was constructed to locate the stem nodes, with an average recognition rate of 93.00% [8]. However, the recognition rate for sugarcane with small differences between the node and internode was low, only 17.00%. Moshashai completed a preliminary study on the identification of sugarcane stem nodes using a threshold segmentation method for a gray image [9]. Lu segmented the HSV colour space of a sugarcane image by the threshold and obtained a composite image by adding the inverse image of the H component image and the S component image [10]. Then, a support vector machine was used to classify and recognize the blocks in the composite image, and the average recognition rate of stem nodes was 93.36%. Huang constructed a rectangular template to move along the horizontal direction of the sugarcane image with a certain step length, then calculated the average gray value on the sugarcane G-B component image [11]. The maximum average gray value was used to determine the position of the stem nodes, but the result was affected by the step length and the width of the template, with a recognition rate of 90.77%. Chen proposed a sugarcane stem node identification method based on the extreme point of a vertical projection function, and the recognition rate of three stem nodes was 95.00% [12]. Although scholars have made significant achievements, there are still some shortcomings in these studies. Under a simple background, the stem node recognition rate of sugarcane images was generally lower than 95.00%. In addition, these studies mainly used traditional image algorithms and gray features to identify sugarcane stem nodes, which rely on artificial segmentation for features.

In addition to traditional image processing algorithms, deep learning has great potential for the task of sugarcane stem node recognition and is already being used in many other areas, such as for facial recognition [13], license plate recognition [14], fruit classification [15], and crop pests and diseases [16,17]. Deep-learning-based methods can automatically learn features from large amounts of data, which means no manual segmentation is required. However, there are few studies on the recognition of sugarcane stem nodes based on deep learning. Li improved the original YOLOv3 network by reducing the number of residual structures, changing the size of the feature map, and reducing the number of anchors, which improved the efficiency of the real-time dynamic recognition of stem nodes [18]. However, the recognition rate was reduced by 2.26%. Therefore, in this paper, deep learning and an image algorithm are combined and an identification and localisation algorithm for sugarcane stem nodes combining YOLOv3 and traditional methods of computer vision is proposed. The algorithm can not only realize fast and accurate identification of sugarcane stem nodes, but also has strong robustness.

The remainder of this paper is organized as follows. In Section 2, we analyse the stem characteristics of the sugarcane variety Xintaitang 22, and then introduce the overall flow to show how to combine YOLOv3 and traditional methods of computer vision to identify sugarcane stem nodes. In Section 3, we propose a new method for obtaining the region of interest in a sugarcane image. Section 4 shows how we built a data set to train the YOLOv3 model and used the model initially to locate stem nodes. Then, we analyse the problems in the experimental results. In Section 5, we propose a new gradient operator and a local threshold determination method, and use them to extract the edges of a sugarcane image. Then, we show how we constructed a stem node localisation algorithm to determine the position of stem nodes. Finally, we verify the effectiveness of the algorithm through experiments, and compare this algorithm with other stem node recognition methods, which proves that the algorithm has the highest recognition rate, fast recognition speed, and high practicability.

## 2. Overall Algorithm Design

### 2.1. Analysis of Cane Stem Characteristics

In this paper, the sugarcane variety Xintaitang 22 was taken as the research object. Figure 1 shows a sugarcane stem. Sugarcane stems are mainly composed of stem nodes, internodes, leaf marks, and buds, as shown in Figure 1. The stem nodes are distributed at approximately 90° to the axis of the sugarcane along the radial direction, and internodes are located between the adjacent stem nodes, on which there are cane buds [19]. There are leaf marks on the stem node and the gray value on both sides is different, so we carried out stem node recognition according to the characteristics of the leaf marks.

### 2.2. Design of a Recognition Process

To further improve the recognition rate of sugarcane stem nodes, an overall identification scheme of the algorithm was designed. Figure 2 shows the flowchart of the identification and localization algorithm of sugarcane stem nodes combining YOLOv3 and traditional methods of computer vision. The recognition process includes three steps: preprocessing, initial identification, and further accurate location. First, the acquired image is preprocessed to correct the sugarcane posture and extract the region of interest. Then, the YOLOv3 network model is used to identify the preprocessed sugarcane image, which initially locates the number and location of the stem nodes. Finally, the improved edge extraction algorithm is used to extract the gradient features of sugarcane, and then a localization algorithm is constructed to determine the position of the sugarcane stem nodes.

## 3. Image Preprocessing

### 3.1. Image Acquisition

Colour images of sugarcane with a black background were collected using a MER133-54GC-P array camera and an M0814-MP2 industrial lens on the DAHENG experimental platform. The Daheng experimental platform is shown in Figure 3. The camera lens was 330 mm away from the cane stem, and the image size was 1280 × 960 Pixels. The image processing program was written in the VS2015 software, which had been configured with OpenCV3.2 to realize the algorithm.

### 3.2. Image Preprocessing

#### 3.2.1. Image Correction

To reduce the noise caused by wax powder between stem nodes, a median filter was used to smooth the original image. The image after noise removal was converted to the HSV colour space for three-channel threshold segmentation. After threshold segmentation, inverse binarization of the segmented binary image was performed to obtain an inverse binary image, as shown in Figure 4a. Judging from the inverse binary image, a large amount of salt-and-pepper noise appeared in the sugarcane background area. The binary image was corroded to remove the highlights on the background. Figure 4b displays the corroded image. Judging from the corroded image, many small holes appeared inside the sugarcane area and breakpoints appeared at the edges of the sugarcane. Therefore, a dilation operation was performed to connect these edges. Figure 4c shows this dilated image.

The “findContours” function in OpenCV was used to find the contour of the image, including the contour of sugarcane and the contour of holes. The function extracted the minimum circumscribed rectangle of each contour, among which the largest circumscribed rectangle was the contour of the sugarcane. The circumscribed rectangle of the contour of sugarcane is shown in Figure 4d. Considering the existence of curved sugarcane, the tilt of sugarcane can cause interference to stem node recognition. Therefore, the original image needed to be transformed and corrected. The sugarcane image was rotated at a certain angle to make the centre axis of the sugarcane parallel to the horizontal axis of the image using an affine transformation. Affine transformation only changes the position of the image relative to the coordinate system, but it does not change the relative position between pixels. The rotation function inputs the sugarcane image and the angle of rotation and outputs the rotated image [20]. The transformation equation for affine transformation is shown in Formula (1):(1)$$\left[\begin{array}{c} x\\ y\\ 1 \end{array}\right]=\left[\begin{array}{ccc} \cos\theta & -\sin\theta & 0\\ \sin\theta & \cos\theta & 0\\ 0 & 0 & 1 \end{array}\right]\left[\begin{array}{c} x_{0}\\ y_{0}\\ 1 \end{array}\right]$$

In Formula (1), ($x_{0},y_{0})$ is the coordinate of a point on the sugarcane image, $\left(x,y\right)$ is the new coordinate of the point after rotation, and θ is the angle of rotation. The angle θ of sugarcane with respect to the horizontal axis can be obtained by the inclination angle of sugarcane’s circumscribed rectangle with respect to the horizontal axis. The coordinates of the four vertexes of the circumscribed rectangle of the sugarcane can be obtained simultaneously. Affine transformation was used to rotate the input image by angle −θ. Figure 4e is the output image after correction.

#### 3.2.2. Extraction of the Region of Interest

To reduce the interference of the background, it is necessary to extract the region of interest of sugarcane. We constructed a 2 × 3 rotation matrix on the polar coordinate system, as shown in Figure 5. In Figure 5, $\left(u,v\right)$ is the coordinate of the vertex of the circumscribed rectangle before rotation, $\left(u_{x},v_{x}\right)$ is the new coordinate of the vertex of the circumscribed rectangle after rotation, θ is the angle of rotation, α is the angle between the horizontal axis and the vertex of the circumscribed rectangle before the rotation, R is the distance between the center point of rotation and the vertex of the circumscribed rectangle before and after rotation, and $\left(col/2,row/2\right)$ is the center point of rotation.

The rotation matrix can be used to calculate the coordinates of the upper left vertex $\left(u_{0},v_{0}\right)$ and the lower right vertex $\left(u_{1},v_{1}\right)$ of the circumscribed rectangle of the sugarcane contour after rotation. The circumscribed rectangle of the sugarcane contour was redrawn through these two points and the circumscribed rectangle was used to obtain an image of the sugarcane region of interest on the sugarcane image after correction. The vertex coordinates of the circumscribed rectangle had the following relationship before and after rotation:(2)$$\left[\begin{array}{c} u_{x}\\ v_{x} \end{array}\right]=\left[\begin{array}{ccc} \cos\theta & \sin\theta & -\frac{col}{2}\times \cos\theta -\frac{row}{2}\times \sin\theta +\frac{col}{2}\\ -\sin\theta & \cos\theta & \frac{col}{2}\times \sin\theta -\frac{row}{2}\times \cos\theta +\frac{row}{2} \end{array}\right]\left[\begin{array}{c} u\\ v\\ 1 \end{array}\right]$$

In Formula (2), $\left(u,v\right)$ is the coordinate of the vertex of the circumscribed rectangle before rotation, $\left(u_{x},v_{x}\right)$ is the new coordinate of the vertex of the circumscribed rectangle after rotation, and θ is the angle of rotation. The image was rotated around the center point $\left(col/2,row/2\right)$, where col and row are the numbers of columns and rows of pixels in the original image, respectively.

The derivation process of the rotation matrix is as follows, in which the coordinates of the vertex of the circumscribed rectangle before and after the rotation can be expressed as Formulas (3) and (4), respectively:(3)$$\left\{\begin{array}{c} u=R\cos\alpha +\frac{col}{2}\\ v=R\sin\alpha +\frac{row}{2} \end{array}\right.$$
(4)$$\left\{\begin{array}{c} u_{x}=R\cos(\alpha +\theta )+\frac{col}{2}\\ v_{x}=R\sin\left(\alpha +\theta \right)+\frac{row}{2} \end{array}\right.$$

We expanded Formula (4) to obtain Formula (5), and then combined Formula (3) with Formula (5) to obtain Formula (6):(5)$$\left\{\begin{array}{c} u_{x}=R\cos\alpha \cos\theta +R\sin\alpha \sin\theta +\frac{col}{2}\\ v_{x}=R\sin\alpha \cos\theta -R\cos\alpha \sin\theta +\frac{row}{2} \end{array}\right.$$
(6)$$\left\{\begin{array}{c} u_{x}=u\cos\theta +v\sin\theta -\frac{col}{2}\cos\theta -\frac{row}{2}\sin\theta +\frac{col}{2}\\ v_{x}=v\cos\theta -u\sin\theta +\frac{col}{2}\sin\theta -\frac{row}{2}\cos\theta +\frac{row}{2} \end{array}\right.$$

Transformation of Formula (6) allowed us to obtain the rotation matrix in Formula (2).

After image preprocessing, the image of the corrected region of interest was obtained. Figure 6 shows the image of the region of interest. The YOLOv3 network model was used to identify the image of the region of interest to preliminarily determine the number and the location of the stem nodes.

## 4. Initial Recognition and Location of Sugarcane Stem Nodes Based on the YOLOv3 Model

As a popular object detection algorithm in deep learning, YOLOv3 was developed from You Only Look Once (YOLO). YOLO was first proposed and applied in the field of target detection in 2015. YOLO regards detection as a regression problem and outputs prediction boxes and classification probabilities [21]. The YOLO algorithm has a fast detection speed and strong versatility. However, for small objects and objects with an unusual aspect ratio, the positioning error is large and the recall rate is low. To solve this problem, YOLO9000 was proposed, which introduces priori boxes of different scales for prediction and can adapt to input objects of different numbers and sizes [22]. Although YOLO9000 improves the detection accuracy, the softmax classifier used in YOLO9000 is not suitable for the detection of overlapping targets. In order to solve the above problems, YOLOv3 was proposed. Compared to other algorithms, YOLOv3 has obvious advantages in speed and precision [23]. Therefore, in this paper, YOLOv3 was applied in the field of sugarcane stem node detection to achieve the rapid and accurate detection of sugarcane stem nodes.

### 4.1. The Data Set

Due to the large size of the original images used in this study, there were non-important image factors, such as image blank. Before creating the data set, all of the original images were preprocessed to obtain corrected images of the sugarcane region of interest. Then, these images were marked with XML, and 750 samples were obtained. Each sample contains about ten stem nodes. Among them, a training set containing 400 samples, a verification set containing 50 samples, and a test set containing 300 samples required for model training were generated. The input image resolution used for model testing was 416 × 416 Pixels. Finally, the YOLOv3 network was constructed to train the dataset, and a YOLOv3 network model was obtained. The captured sugarcane images are shown in Figure 7.

### 4.2. Evaluation Criteria

The precision rate, recall rate, harmonic mean, and image recognition speed were used as evaluation criteria for the model. The precision rate P is the proportion of the actual positive classes in the predicted positive classes, and the recall rate R is the proportion of the positive classes successfully predicted in the actual positive classes. “Average Recognition Time/s” is the time taken to recognize all stem nodes in a single image.

Because the precision rate *P* and recall rate *R* cannot reach the optimal value at the same time, the harmonic mean *F* was also used as an evaluation index. The calculation method is shown in Formulas (7)–(9), where *TP* is the number of samples that are actually positive classes and are successfully predicted as positive classes, namely, the real positive classes; *FP* is the number of samples that are negative classes but predicted to be positive classes, namely, the false-positive classes; *FN* is the number of samples that are actual positive classes but predicted to be negative classes, namely, the false-negative classes.
(7)$$P=\frac{TP}{TP+FP}$$
(8)$$R=\frac{TP}{TP+FN}$$
(9)$$F=\frac{P\times R\times 2}{P+R}$$

### 4.3. Analysis of the Model Performance

The model performance was tested using 300 sugarcane images in the test set. These images contained 615 stem nodes, and the model recognized 631 stem nodes, including 615 TP, 16 FP, and 0 FN.

As shown in Table 1, the recall rate of this model reached 100%, which means all of the stem nodes on the sugarcane had been identified. The precision rate was 97.46%. Figure 7 shows the output image of the YOLOv3 network model. Figure 8a shows the output results after the correct recognition of sugarcane images with three stem nodes. The model output three prediction boxes, each of which had a confidence degree. The coordinates of the upper left corner of the prediction box and the length and width of the prediction box could be used to initially determine the position of sugarcane stem nodes.

The main reason for the relatively low precision is that a single stem node was repeatedly identified during the test. As shown in Figure 8b, the model output two prediction boxes at the position of the middle stem node, which means that the model repeatedly predicted the middle stem node twice. During the test, the position of the non-stem node was also wrongly predicted as the stem node. As shown in Figure 8c, the leaf mark of sugarcane was exactly at the boundary of the image. The model identified the stem node without the leaf mark as the stem node. The harmonic mean was 98.72%, and the average recognition time was 0.17 s.

There were two types of errors in the recognition results of the YOLOv3 model, leading to a relatively low recognition rate. In order to further determine the location of the sugarcane stem nodes and improve the accuracy of stem node recognition, we used the improved edge extraction algorithm and the localization algorithm for follow-up processing of the recognition results of the YOLOv3 model to correct the errors in YOLOv3 recognition.

## 5. Accurate Recognition and Location of Sugarcane Stem Nodes Based on the Improved Edge Extraction Algorithm and the Localization Algorithm

### 5.1. Canny Algorithm and Its Defects

The traditional Canny edge detection algorithm uses the first-order differential to extract edges, and the detection process can be summarized as follows: (1) Gaussian filtering is used to remove noise; (2) an appropriate gradient operator is selected to calculate the amplitude and direction of the gradient, with the Sobel gradient operator commonly being used in the Canny algorithm; (3) non-maximum suppression of the gradient amplitude is carried out, for which the gray value corresponding to the non-maximum point is set to 0 to suppress the non-edge point; (4) the double-threshold algorithm is used to detect and connect edges, including a high threshold and a low threshold. The edge image is obtained according to the high threshold, and only the edge points with continuous paths are retained according to the low threshold and connect to the edge image obtained by the high threshold [24].

During the experiment, the results of the three-channel separation of RGB images were observed, and it was found that the G component and B component had uniform grayscale at the stem node and interstem position, while the R component image had a narrow highlight band at the stem node with strong gray-dark contrast with the interstem, and the obvious difference in grayscale values between the stem node and interstem was beneficial to the stem node identification. Therefore, R component is selected as the input image.

Figure 9 shows the edge detection results for a sugarcane R component image which is from an RGB image with different operators, in which Figure 9a is the sugarcane R component image and Figure 9b is the edge detection result of the Canny operator. The Canny operator was not susceptible to noise interference and retained relatively weak edges. However, the difference between stem nodes and internodes was too small for identifying stem nodes from edge images. In addition, the threshold value of the traditional Canny algorithm is globally fixed and determined mainly according to manual experience. Therefore, it is difficult to obtain an optimal threshold value.

### 5.2. Improved Gradient Extraction Algorithm

#### 5.2.1. Improved Gradient Operator

After extracting the R component image from the sugarcane image and filtering it, a new gradient operator was used to calculate the gray gradient of the sugarcane image. Formula (10) shows the calculation equation of the gradient operator G:(10)$$G=\left[\begin{array}{ccc} -1 & 0 & 1\\ -1.5 & 0 & 1.5\\ -1 & 0 & 1 \end{array}\right]*A$$

The new operator contained a 3 × 3 matrix, convolved with the image to obtain the gradient value in the horizontal direction of sugarcane. A represents the R component image of sugarcane, and G represents the gradient value of the image after edge detection.

Formula (11) is the expansion of the gradient operator:(11)$$G=f\left(x+1,y-1\right)+1.5\times f\;\left(x+1,y\right)+f\left(x+1,y+1\right)\;-f\left(x-1,y-1\right)-1.5\times f\left(x-1,y\right)-f\left(x-1,y+1\right)$$

In Formula (11), $f\left(x,y\right)$ represents the gray value of point $\left(x,y\right)$ of the sugarcane image, and G is the gradient value of the point $\left(x,y\right)$ of the sugarcane image. The gradient value of each point in the image was obtained by subtracting the gray value of the adjacent points in the horizontal direction of the pixel center point. Figure 9c shows the edge detection results of the new gradient operator. Compared to the Canny operator, the new operator was able to enhance the gradient value of the stem node in the sugarcane image and eliminate the weak edge between stem nodes, highlighting the small difference between the stem node and internode. Therefore, the new operator is conducive to stem node identification.

#### 5.2.2. Local Threshold

After edge detection by the gradient operator, binarization is required. For different images, using a fixed threshold value will lead to the failure to find the real edge accurately. Therefore, a local threshold determination method is proposed in this paper. First, the image was traversed to calculate the average gradient value of the image. Then, the prediction boxes obtained by the YOLOv3 model were utilized. There was a high probability of sugarcane stem nodes appearing in the prediction boxes. The gradient value of the stem node was larger, and the gradient value between the stem nodes was generally low. The gradient value of the region influenced by wax powder and broken leaves. To reduce this noise, the threshold value should be as large as possible. Therefore, it was necessary to eliminate all points in the prediction box whose gradient value was lower than the average of the image gradient value. The average gradient value of the remaining points in the prediction boxes was calculated to obtain the local threshold, which was used to binarize the image. Figure 10 is the flow chart of local threshold determination algorithm.

In the Figure 10, Avg is the gradient mean of the edge detection image, Box(*x*,*j*) is the gradient value of point (*i*,*j*) in the prediction box containing stem nodes, Loc is the local threshold, *i_max_* is the total number of pixels in the prediction box, *n* is the number of pixels in the prediction box whose gradient value is greater than the gradient mean. Sum is the sum of the gradient values of the points in the prediction box whose gradient value is greater than the gradient mean. Firstly, the edge detection image is loaded, the image is traversed to calculate the image gradient mean, and then the prediction box output by YOLOv3 network model is used to find that there is a high probability of stem nodes in the prediction box, in which the gradient value of stem nodes is large, and the gradient value of interstem is small. The gradient value of partial stem nodes will mutate due to the interference of wax powder and broken leaves. To eliminate this noise, the threshold should be as large as possible. Therefore, all the points with gradient values lower than the image mean in the prediction box were eliminated, and the gradient mean of the remaining points in the prediction box was calculated to obtain the local threshold.

Formula (12) shows the binarization process:(12)$$q\left(x,y\right)=\left\{\begin{array}{c} 255,G\left(x,y\right)\geq T\\ 0,\;G\left(x,y\right)<T \end{array}\right.$$

In Formula (12), $G\left(x,y\right)$ represents the gradient value of a point on the edge detection image, and $q\left(x,y\right)$ represents the gradient value of the point after binarizing the image by the local threshold.

The binarization results are shown in Figure 11. After binarization, the number of noise points between stem nodes was significantly reduced. Therefore, it can be concluded that local threshold binarization has a good denoising effect.

### 5.3. Localisation Algorithm for Stem Nodes

After binarization of the gradient image, the gradient value of the stem node area was the largest, while the gradient value of the non-stem node area was relatively small, therefore the stem node positioning process is as follows.

①The positions of the prediction boxes are taken as the detection regions of stem nodes. If the distance between the centre of the detection regions is less than 50 pixels, the two prediction boxes are combined as a new detection region of stem nodes, such as Figure 8b;②The gradient values of each column in the detection area are summed and the position of the maximum value is taken as the position of the suspected stem node;③If the sum of the column gradient values of the position of the suspected stem node is greater than or equal to 200, it will be judged as a stem node and marked on the original image. Otherwise, it will be judged as a non-stem node.

Step ① is used in the case of repeated recognition of the YOLOv3 model. The 50 in step ② refers to experimental data because the maximum centre distance of the repeated prediction boxes output by the YOLOv3 model is less than 50 pixels. The threshold judgment for the sum of the gradient values of a column in step ③ is mainly for cases where the YOLOv3 model incorrectly predicts the non-stem node as the stem node. By analysing the column gradient values at the position of the stem node of the sample image, the minimum value of the column gradient values at the position of the stem node was 200, which was used as the threshold value for judging the stem node.

## 6. Experimental Results and Discussion

### 6.1. Analysis of the Identification Results

The algorithm combining YOLOv3 and traditional methods of computer vision was tested using 300 images in the test set, which contained 615 stem nodes in total. The algorithm recognized 617 stem nodes, including 615 TP, 2 FP, and 0 FN. The precision rate, recall rate, and harmonic mean were 99.68%, 100%, and 99.84%, respectively. A comparison of the combined algorithm and YOLOv3 algorithm is shown in Table 2. The experimental results show that the precision rate of the combined algorithm was improved by 2.28% and the harmonic mean by 1.13%.

Compared to the output results of the YOLOv3 network model, the experimental results show that the algorithm that combined YOLOv3 and traditional methods of computer vision can effectively improve the accuracy of sugarcane stem node recognition. Figure 12 shows the output results of the algorithm that combined YOLOv3 and traditional methods of computer vision. Figure 12a shows the output results of the image of three stem nodes, Figure 12b shows the output results after improvement of repeated recognition, and Figure 12c shows the output result after improvement of non-stem node recognition. The “sum of column gradient” in Figure 12 indicates the sum of the gradient vector values of a single column of the image at the location of this pixel, “column/pixel” indicates the horizontal pixel value of the image.

### 6.2. Comparison with Other Methods

Table 3 shows a comparison of the average recognition rate and average recognition time of various recognition methods. According to the results, after comparison, the recognition algorithm for the sugarcane stem nodes based on YOLOv3 and traditional methods of computer vision in this paper had the highest recognition rate—except for the recognition speed found in [18], which was only 0.0287 s, the recognition speed of the algorithm in this paper was the fastest. The results show that the algorithm in this paper has good real-time performance.

## 7. Conclusions

Aiming at the obvious characteristics of sugarcane leaf marks, leaf marks are generally used to identify sugarcane stem nodes. In this paper we proposed a stem node identification and localization algorithm combining YOLOv3 and traditional methods of computer vision. The major conclusions from the conducted study are as follows:(1)A new gradient operator was used to extract the edge of a sugarcane R component image. Compared to the Canny operator, the experimental results show that the new operator is better. The stem node has a strong margin and the margin of the internode is thinner, which can greatly highlight slight differences between the stem node and the internode;(2)A local threshold determination method was proposed, which removes pixels whose gradient values in the prediction box are lower than the average value of the image. Then, the average gradient value of the remaining points in the prediction box is calculated to obtain the local threshold, which is used as the threshold of binarization of the edge detection image. The experimental results show that the noise between stem nodes is obviously reduced after binarization, which means the local threshold binarization has a good denoising effect;(3)We used polar coordinates to derive a rotation matrix. The rotation matrix is used to calculate the coordinates of the upper left vertex and the lower right vertex of the circumscribed rectangle of the sugarcane contour after rotation. The circumscribed rectangle of the sugarcane contour is redrawn through two points to obtain a sugarcane image of the region of interest after rotation, eliminating the influence of the background on image recognition; and(4)We proposed an identification and localization algorithm for sugarcane stem nodes combining a YOLOv3 network and traditional methods of computer vision. The experimental results show that the precision rate of the identification algorithm for sugarcane stem nodes proposed in this paper was 99.68%, the recall rate was 100%, and the harmonic mean was 99.84%. Compared to the original network, the precision rate and harmonic mean were improved by 2.28% and 1.13%, respectively.

Although the improved model can accurately detect the location of sugarcane stem nodes, it lacks the detection of whether the cane seeds are diseased. In the future, we can consider using means such as images or near-infrared spectroscopy to identify diseased sugarcane by analysing its stem, colour or internal composition, and include a rejection device for such unqualified cane seeds in the seed cutting equipment. At the same time, the recognition speed determines the efficiency of mechanical seed cutting, so the research on the recognition rate will be increased in the next step.

## Figures and Tables

**Figure 1 sensors-22-08266-f001:**
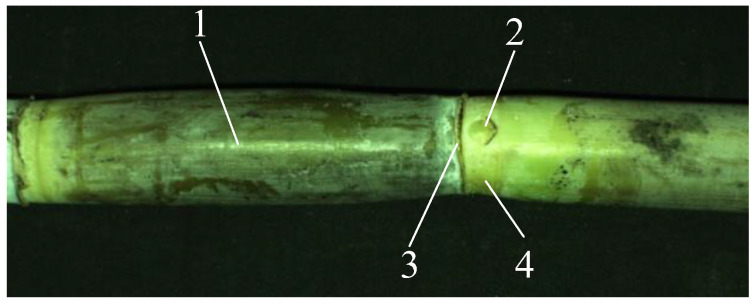
Sugarcane stem. (1) Internode, (2) cane bud, (3) leaf mark, and (4) stem node.

**Figure 2 sensors-22-08266-f002:**
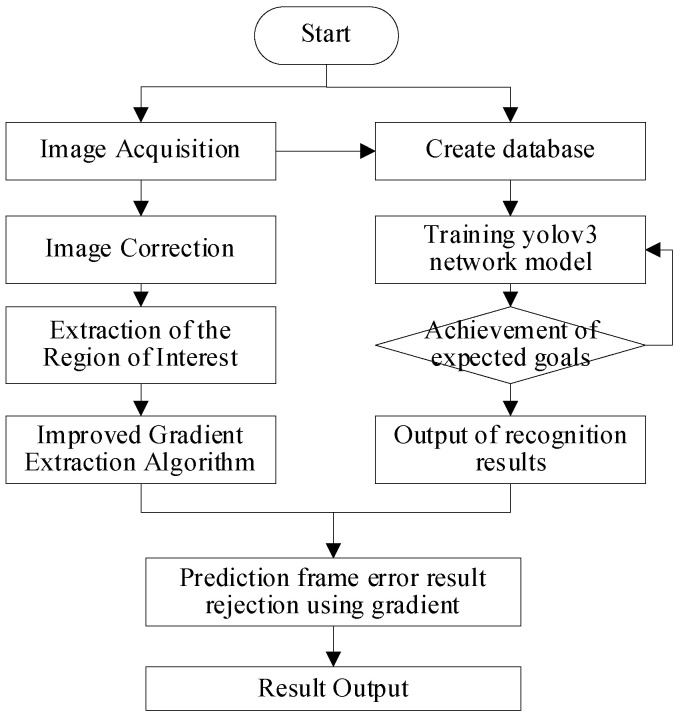
Flowchart of the identification and localization algorithm of sugarcane stem nodes combining YOLOv3 and traditional methods of computer vision.

**Figure 3 sensors-22-08266-f003:**
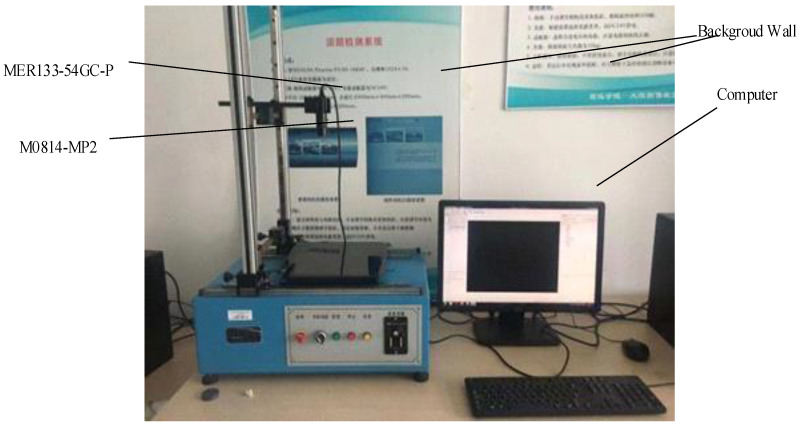
DAHENG experimental platform.

**Figure 4 sensors-22-08266-f004:**
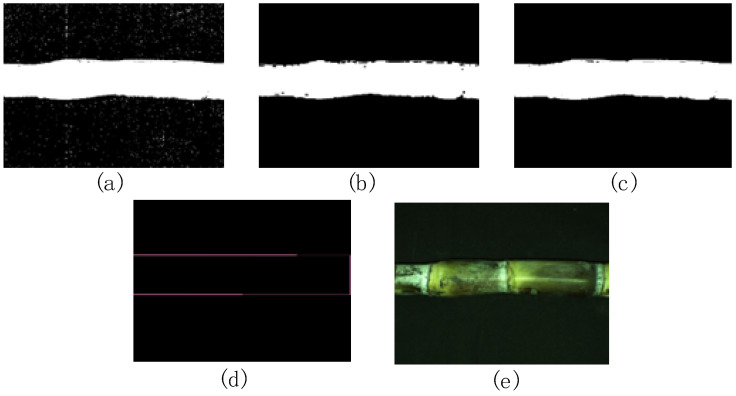
Preprocessing of sugarcane images: (**a**) inverse binary image, (**b**) corroded image, (**c**) dilated image, (**d**) circumscribed rectangle, and (**e**) image after correction.

**Figure 5 sensors-22-08266-f005:**
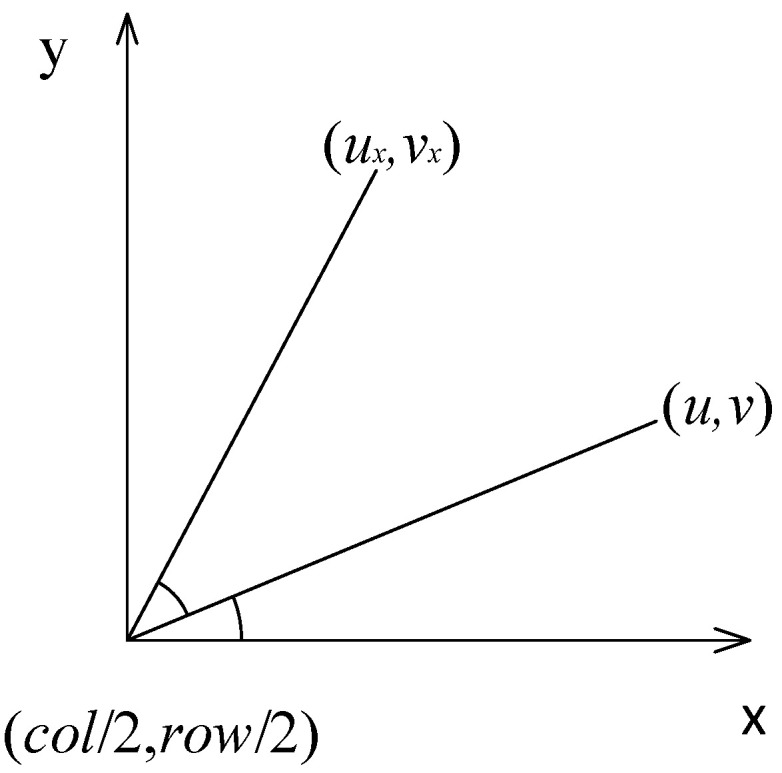
Polar coordinate system.

**Figure 6 sensors-22-08266-f006:**
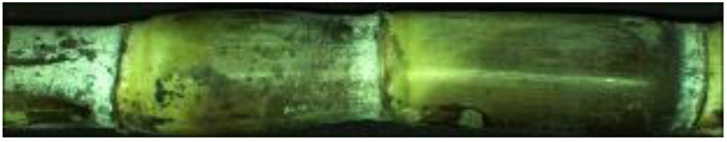
Image of the region of interest.

**Figure 7 sensors-22-08266-f007:**
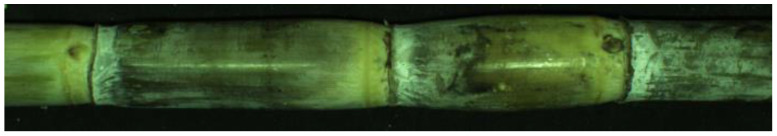
Sugar cane images.

**Figure 8 sensors-22-08266-f008:**
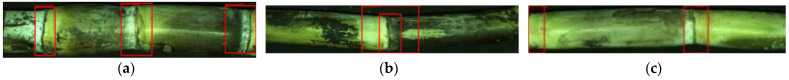
Output of the YOLOv3 network model: (**a**) two stem nodes, (**b**) repeated recognition, (**c**) non-stem-node recognition into stem node.

**Figure 9 sensors-22-08266-f009:**
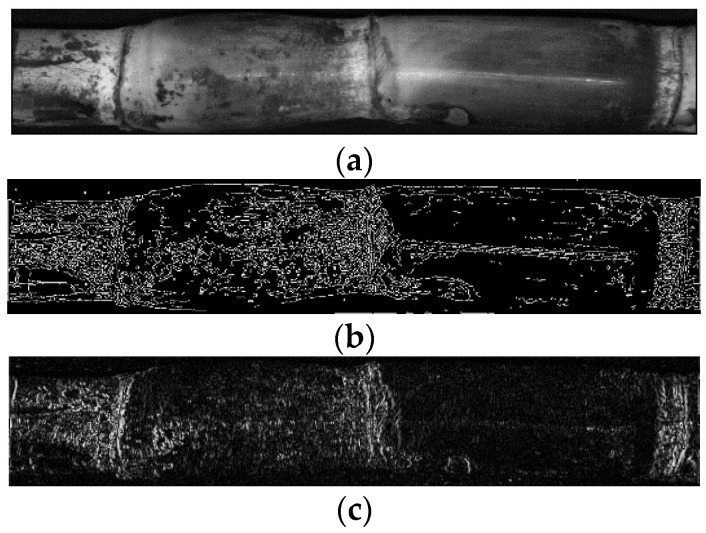
Edge detection results for the sugarcane R component image with different operators: (**a**) sugarcane R component image, (**b**) edge detection result of Canny operator, and (**c**) edge detection result of the new gradient operator.

**Figure 10 sensors-22-08266-f010:**
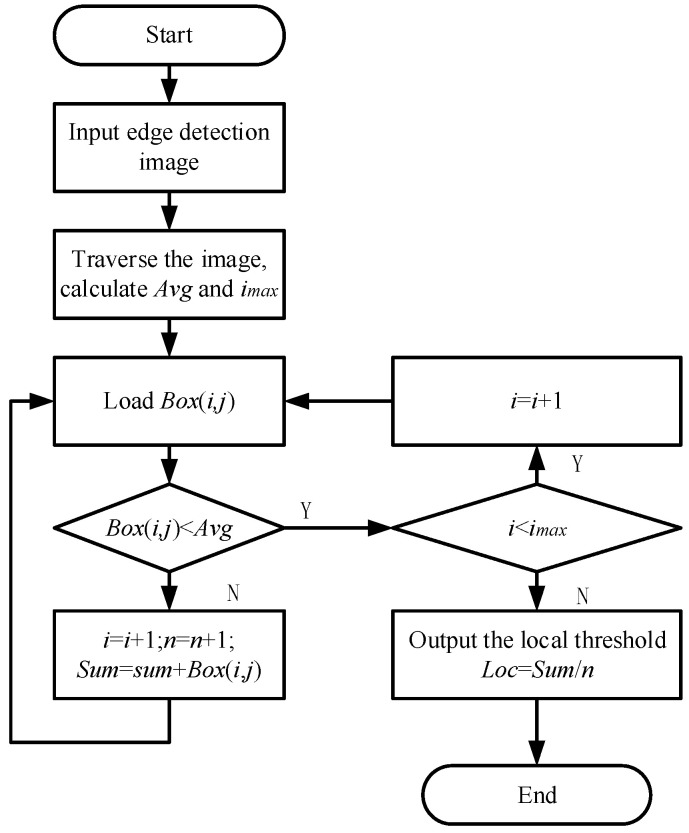
Flow chart of local threshold determination algorithm.

**Figure 11 sensors-22-08266-f011:**
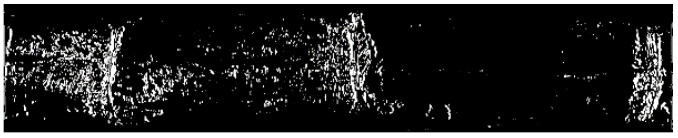
Binarization results of the gradient image.

**Figure 12 sensors-22-08266-f012:**
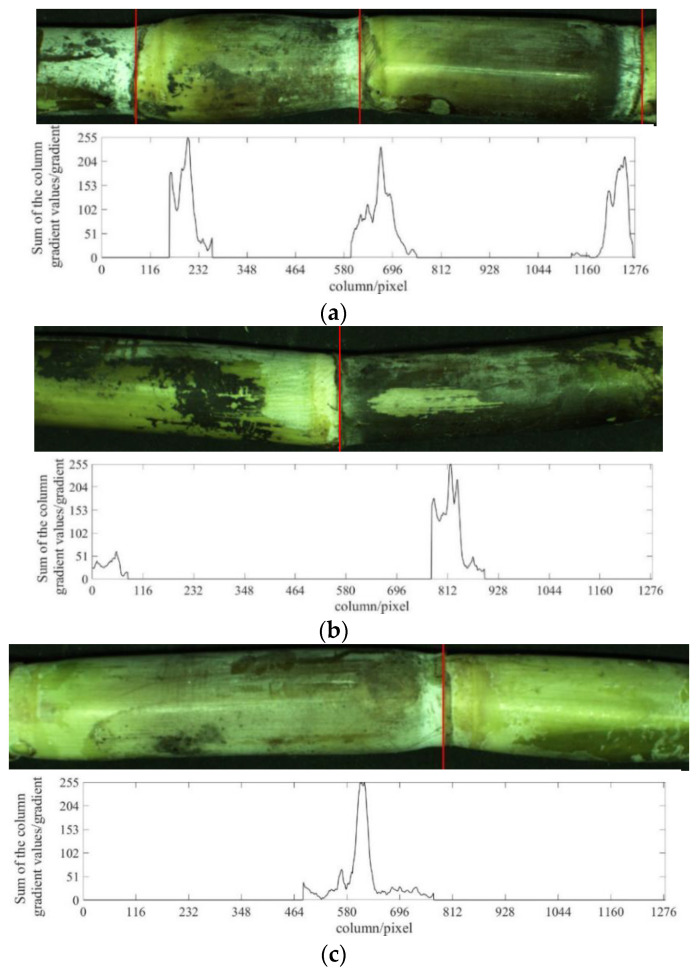
The output results of the algorithm combining YOLOv3 and traditional methods of computer vision: (**a**) three stem nodes, (**b**) repeated recognition, and (**c**) non-stem node recognition into stem node.

**Table 1 sensors-22-08266-t001:** Statistics of the detection results.

Model	TP	FP	FN	Precision (P)	Recall (R)	Harmonic Mean (F)	Average Recognition Time/s
YOLOv3	615	16	0	97.46	100	98.72	0.17

**Table 2 sensors-22-08266-t002:** Comparison of the combined algorithm and the YOLOv3 algorithm.

Model	TP	FP	FN	Precision (P)	Recall (R)	Harmonic Mean (F)	Average Recognition Time/s
Improved algorithm	615	2	0	99.68	100	99.84	0.415
YOLOv3	615	16	0	97.46	100	98.72	0.17

**Table 3 sensors-22-08266-t003:** Comparison of different recognition methods.

Methods	Number of Samples	Methods Detail	Average Recognition Rate/%	Average Recognition Time/s
Zhou et al. [8]	119	Search potential node positions in the gradient feature vector.	93.00	0.539
Lu et al. [10]	3200	Clustering analysis was introduced to identify sugarcane nodes blocks which were got by support vector machine.	93.36	0.76
Huang et al. [11]	130	The corresponding position of maximum average grey value determine the position of sugarcane nodes.	90.77	0.481
Li et al. [18]	12,000	Improve YOLOv3 network by reduce the residual junction formed and number of anchors.	90.38	0.0287
The algorithm in this paper	750		99.84	0.415

## Data Availability

Not applicable.

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
