# Peer review of "Identification and Localisation Algorithm for Sugarcane Stem Nodes by Combining YOLOv3 and Traditional Methods of Computer Vision"

_sensors, 2022, doi:10.3390/s22218266_

Round 1

Reviewer 1 Report

Dear authors,

Kindly refer to attachment for feedback.

Regards

Author Response

Dear Editors and Reviewers:

Thank you for your letter and for the reviewers' comments concerning our manuscript entitled " Identification and Location Algorithm for Sugarcane Stem Nodes Combining YOLOv3 and Machine Vision" (sensors-1854804). Those comments are all valuable and very helpful for revising and improving our paper, as well as the important guiding significance to our researches. We have studied comments carefully and have made correction which we hope meet with approval.

Revised portion are marked in red in the paper. The main corrections in the paper and the responds to the reviewer's comments are as flowing:

Responds to the reviewer's comments:

Reviewer #1:

  1. Check for spelling error, such as “inage blank” in line 223

Response:

We are very sorry for our incorrect writing. We have carefully checked for spelling errors and changed "inage blank" to "image blank" in line 223;

  1. Please clarify in Section 4.1 whether the 750 samples are images of only one sugarcane stem node.

Response:

We have made correction according to the Reviewer’s comments. Each sample contains about 10 stem nodes, which have been added to the description in line 224 and marked in red;

  1. The statement "750 training samples" is misleading.

Response:

            We are very sorry for our negligence. The phrase "750 training samples" is unclear and should be changed to "750 samples".

  1. Is this the average recognition time/s for a single image?

Response:

            Yes, "Average Recognition Time/s" is the time taken to recognize all stems in a single image, description added in line 235.

  1. R component is not defined.

Response:

            We are very sorry for our negligence. The description of the R component has been added in line 285 and marked in red.

  1. Kindly provide a simple diagram to support the explanation.

Response:

It is really true as Reviewer suggested that Figure 7 (b) matches the description, and we add a reference to Figure 7 (b).

  1. Figure 10. Its best to explain using the same image rather than different image when explaining the effect of the added improvements. It is also a good idea to further explain the sum of column gradient values/gradient vs. column/pixel charts.

Response:

As Reviewer suggested that the picture used to show the improved effect is the same as the unimproved picture in Figure 7;

And the explanation of "sum of column gradient values/gradients" and "column/piexl" is added in line 373. " in line 373.8.

  1. Table 3. Best to add one more column that describes a brief detail of their recognition method.

Response:

We have made correction according to the Reviewer’s comments.We added a column describing short details, see Table3.

  1. Future work and limitation is missing.

Response:

            As Reviewer suggested that we add future work and limitation in Conclusion.

Reviewer 2 Report

In this manuscript, the authors proposed a so-called "YOLOv3 and machine vision" to solve the challeging  problem of detecting stem nodes quickly and accurately. The novelty is limited, since YOLOv3 performs well in some PR tasks and it is not clear what is the specific meaning by term "machine vision". The main weakness of the method is that no comparing results have been listed, no ablation study is involved. So, the novelty is quite limited by showing some data experimental results on  detecting stem by YOLOv3 and machine vision.

Author Response

Dear Editors and Reviewers:

Thank you for your letter and for the reviewers' comments concerning our manuscript entitled " Identification and Location Algorithm for Sugarcane Stem Nodes Combining YOLOv3 and Machine Vision" (sensors-1854804). 

I would like to explain something to you. the YOLOv3 network does perform well in some PR tasks, but as shown in Figure 7, Figure b, Figure c. there are two situations that can affect PR. So by combining with opencv to eliminate this part of the effect, and finally in the experiments in subsection 6.1, it was shown by comparison experiments that this approach does work. And in the end there is an advantage compared with other methods in the same field.

Reviewer 3 Report

Authors propose a Yolo V3 + gradient operator for sugarcane stem node identification and localisation algorithm. The technical novelty is rather limited, and it is not clear why a gradient operator is needed. It is rather unconventional nowadays to compare with Canny, instead of having an end-to-end deep learning system. Handcrafting these features need to be compared against end-to-end deep learning system to demonstrate the superiority of such a method.

In addition, I am not sure how the authors create a divide between YoLo and Machine Vision. Isn't YoLo a machine/computer vision algoirthm based on deep learning? I think the authors refer to more traditional methods of computer vision, such as Canny, when they mention "machine vision"

Lines 43/44 need rephrasing; what is meant by "home and abroad"? The paper should be written for international audience so if there is any reason to primarily focus on one country and then try to generalise to the global setting then that needs to be made clear.

Introduction needs some revising; second paragraph is too long and it jumps from one topic to another. Perhaps it would be helpful to first define the topic, then define the problem, then provide the context and background with references etc., then list the contributions of this work and then finish off with the organisation of the remainder of the paper (as you have done)

Figures 2, and 3 are not fit for purpose for such a paper; they are too simplistic and uninformative. 

By "location algorithm" do you mean "localisation algorithm", as in using an algorithm to localise something in an image?

Finally, conclusion is written more like a summary describing findings rather than providing concluding remarks, main findings, limitations, importance of the results, and future work.

Author Response

Dear Editors and Reviewers:

Thank you for your letter and for the reviewers' comments concerning our manuscript entitled " Identification and Location Algorithm for Sugarcane Stem Nodes Combining YOLOv3 and Machine Vision" (sensors-1854804). Those comments are all valuable and very helpful for revising and improving our paper, as well as the important guiding significance to our researches. We have studied comments carefully and have made correction which we hope meet with approval.

Revised portion are marked in red in the paper. The main corrections in the paper and the responds to the reviewer's comments are as flowing:

Responds to the reviewer's comments:

Reviewer #3:

  1. Handcrafting these features need to be compared against end-to-end deep learning system to demonstrate the superiority of such a method

Response:

The superiority of the new operator is demonstrated in the comparative tests in Figure 8, sub-section 5.2.1, b and c, and the superiority of the new algorithm over the YOLOv3 algorithm in all metrics is demonstrated by data in Table 2, sub-section 6.1

  1. I think the authors refer to more traditional methods of computer vision, such as Canny, when they mention "machine vision".

Response:

I am sorry that there is an error in the exposition, it should be "traditional methods of computer vision".;

  1. what is meant by "home and abroad"?

Response:

      There is no specific focus on a particular country, so replace "domestic and international" with "worldwide".

  1. Introduction needs some revising

Response:

       The introduction has been reworked as suggested by the reviewers.

  1. Figures 2, and 3 are not fit for purpose for such a paper; they are too simplistic and uninformative.

Response:

       I has been redrawn in detail for Figure 2, and Figure 3 shows the actual picture collection scenario, which does not involve complex scenes, and therefore the pictures are simpler..

  1. By "location algorithm" do you mean "localisation algorithm", as in using an algorithm to localise something in an image?.

Response:

Using algorithms to locate the stem nodes of sugarcane in order to locate the position of the seed buds of sugarcane and to achieve intelligent cutting of seeds by preventing injury to the buds.I has been changed from "location algorithm" to " localisation algorithm".

  1. Conclusion is written more like a summary describing findings rather than providing concluding remarks, main findings, limitations, importance of the results, and future work

Response:

The conclusions have been refreshed to add limitations, results, and future work.

Reviewer 4 Report

please check the pdf.

Author Response

Dear Editors and Reviewers:

Thank you for your letter and for the reviewers' comments concerning our manuscript entitled " Identification and Location Algorithm for Sugarcane Stem Nodes Combining YOLOv3 and Machine Vision" (sensors-1854804). Those comments are all valuable and very helpful for revising and improving our paper, as well as the important guiding significance to our researches. We have studied comments carefully and have made correction which we hope meet with approval.

Revised portion are marked in red in the paper. The main corrections in the paper and the responds to the reviewer's comments are as flowing:

Responds to the reviewer's comments:

Reviewer #1:

  1. Why are some sentences marked in red ?Revised,or has other special meaning.

Response:

The red part indicates the department that was revised in the first round of amendments.

  1. The title and body of Figure 5/6/… are split into separate pages.

Response:

I am sorry that I take a mistake. I have adjusted the position of the diagram.

  1. In section 4.1,more instances of the dataset need to be given.In particular,the object of the category being annotated.

Response:

In section4.1,I have been add picture of dataset.

  1. In equation(10),why 1.5,is there any special meaning here?The author needs to give more clarification,from what literature or experments?

Response:

In the selection of 1.5, a comparative test is done. Sobel operator, Sobel horizontal gradient operator, Prewitt operator and Prewitt horizontal gradient operator are used to measure the edge of the binarized image respectively. The last two horizontal operators have a good effect, and 1.5 is obtained by weighting them. This article does not focus on this, so there is no explanation.

  1. Overall,the author’s experiments are not sufficient.if real-time is a focus,more lightweight models need tobe compared and analyzed.E.g,tiny-yolov3,tiny-yolov4,etc.

Response:

This paper focuses on the accuracy rate and makes a horizontal comparison in 6.1. Real-time performance is only for reference, not the focus.

Round 2

Reviewer 1 Report

1. Section 5.1, line 285. Kindly justify as to why R component is chosen rather than G or B.

2. Kindly increase the font size of Figure 10 x and y axis label. Its too difficult to see without zooming in multiple times.

3. Table 3 can be further improved by stating the metrics presented in the Table. For example, the algorithm in this paper reported a 99.84% Harmonic Mean. What about the other methods? Are they also uses Harmonic Mean/F-Measure as their main metrics?

Author Response

Dear Editors and Reviewers:

Thank you for your letter and for the reviewers' comments concerning our manuscript entitled " Identification and Location Algorithm for Sugarcane Stem Nodes Combining YOLOv3 and Machine Vision" (sensors-1854804). Those comments are all valuable and very helpful for revising and improving our paper, as well as the important guiding significance to our researches. We have studied comments carefully and have made correction which we hope meet with approval.

Revised portion are marked in red in the paper. The main corrections in the paper and the responds to the reviewer's comments are as flowing:

Responds to the reviewer's comments:

Reviewer #1:

  1. Section 5.1, line 285. Kindly justify as to why R component is chosen rather than G or B.

Response:

During the experiment, the results of the three-channel separation of RGB images were observed, and it was found that the G component and B component had uniform grayscale at the stem node and interstem position, while the R component image had a narrow highlight band at the stem node with strong gray-dark contrast with the interstem, and the obvious difference in grayscale values between the stem node and interstem was beneficial to the stem node identification. Therefore, R component is selected as the input image.

  1. Kindly increase the font size of Figure 10 x and y axis label. Its too difficult to see without zooming in multiple times.

Response:

I am sorry that I take a mistake.I have been increase the font size of Figure 10.

  1. Table 3 can be further improved by stating the metrics presented in the Table. For example, the algorithm in this paper reported a 99.84% Harmonic Mean. What about the other methods? Are they also uses Harmonic Mean/F-Measure as their main metrics?

Response:

The focus of this paper is mainly on the improvement of the recognition rate, so it is also based on this parameter to compare with other methods

Reviewer 2 Report

All my comments have been addressed, I recommend to accept it for publication.

Author Response

Thank you

Reviewer 3 Report

I think the paper's novelty is rather limited, requiring a comprehensive rethinking of its design and methods proposed. 

Author Response

A stem node recognition model based on the improved YOLOv3 network was designed to initially locate the stem node position. For the two types of errors in the recognition results of the YOLOv3 network model, an improved edge extraction algorithm and a stem node localization algorithm were proposed to accurately identify and locate the sugarcane stem nodes, and offline tests showed that the average recognition rate was 99.84% and the recognition time was 415ms

Reviewer 4 Report

The paper has some engineering and scientific interest, but the introduction, data, and experiments mentioned in the previous round have barely been revised.

Author Response

Dear reviewer, please find the specific modification reply in the attachment.

Round 3

Reviewer 3 Report

The authors have addressed all the major concerns raised in my original review therefore I am happy for it to be accepted.

Author Response

  • Thank you for your review

Reviewer 4 Report

The paper has some engineering and scientific interest, but the introduction, data, and experiments mentioned in the previous round have barely been revised.

Author Response

Dear Reviewers,

Details of the last review comments revision are in the cover letter.
